# Looking beyond the Shoal: Fish Welfare as an Individual Attribute

**DOI:** 10.3390/ani12192592

**Published:** 2022-09-28

**Authors:** Lauri Torgerson-White, Walter Sánchez-Suárez

**Affiliations:** 1Department of Research, Farm Sanctuary, Watkins Glen, NY 14891, USA; 2Department of Research, Mercy For Animals, Los Angeles, CA 90046, USA

**Keywords:** animal welfare, fish welfare, individuality, emotion, preferences, personality, cognition, animal welfare assessment

## Abstract

**Simple Summary:**

The fish farming industry is characterized by settings where large numbers of fishes are raised together at high stocking densities, effectively obliterating the individual. Given that animal welfare is an individual attribute that refers to how an animal experiences her world, it follows that ensuring good welfare for the different individuals is difficult in fish farms. In this paper we review evidence supporting the notion that fishes are individuals and fish welfare should thus also be considered at the individual level, examine the ways that animal welfare is assessed in fish farms, evaluate these practices in light of individualized terrestrial animal welfare assessment methods, and make recommendations regarding research that could lead to a better understanding of how to provide each individual fish with good welfare in captivity.

**Abstract:**

Welfare is an individual attribute. In general, providing captive nonhuman animals with conditions conducive to good welfare is an idea more easily applied when dealing with few individuals. However, this becomes much harder—if not impossible—under farming conditions that may imply high numbers of animals living in large group sizes. Although this is a problem inherent to intensive animal farming, it is possibly best exemplified in fish farming, for these practices often rely on extremely high numbers. In this paper we review evidence supporting the notion that fishes are individuals and fish welfare should thus also be considered at the individual level, examine the current state of welfare assessment in the aquaculture industry, evaluate these practices in light of individualized terrestrial animal welfare assessment methods, and make recommendations regarding research that could lead to a better understanding of how to provide each individual fish with good welfare in captivity.

## 1. Introduction

Animal welfare science emerged as a formal scientific discipline after Ruth Harrison’s *Animal Machines* criticized confinement systems used in industrial animal agriculture, in part leading to the concept of the Five Freedoms [1]. This history, rooted in terrestrial animal agriculture, makes clear the utility of examining welfare assessment techniques, inputs, and outcomes across a variety of types of animal husbandry systems, and integrating and applying that knowledge to optimize welfare. Indeed, without these cross-discipline applications of terrestrial farm animal welfare science, zoo animal welfare science and laboratory animal welfare science may not exist. Given the relative numbers of animals involved, it is not surprising that efforts to provide good welfare at the level of the individual animal in zoos and laboratories are more successful than are similar efforts in intensively farmed animals. In light of the relative paucity of fish welfare research in general—especially considering that farmed fishes span hundreds of species living in different environments and having different needs [2], and the nascence and rapid expansion of industrial aquaculture, fish welfare scientists can learn from those disciplines that have made more progress with elevating individual welfare.

In many species, even within the same population, individuals show consistent differences in how they react, behaviorally and physiologically, to environmental stimuli, resulting in a range of welfare states in response to the same stimulus. Such differences are sometimes described as reflecting individual personalities and may be expressed as variation in coping strategy [3,4], in preferences [5,6] and in tendency toward optimism or pessimism [7,8,9,10]. Given the general consensus among animal welfare scientists that welfare is, first and foremost, an attribute of an individual, [1,11,12] not a flock, herd, or shoal, some argue that welfare should thus be measured primarily at the level of the individual [13,14,15,16,17,18,19]. For the sake of clarity and building on previous definitions [20,21,22,23,24,25,26], we regard welfare as the emotional experience of an individual that results from that fish’s interactions with her environment over time, existing on a continuum from negative to positive.

As an intensive type of vertebrate farming, the aquaculture industry is currently ill-equipped to properly address the needs of all individuals. First, consider the sheer number of animals being farmed. While the most recent report from the Food and Agriculture Organization of the United Nations reports aquaculture production in tonnes rather than individual fishes [27], estimates suggest that in 2017, between 51 and 167 billion fishes were farmed for food [28]. Although some fish farmers regularly monitor the condition and behavior of the fishes they farm, with group numbers upwards of 100,000 in an underwater enclosure, they clearly cannot observe and keep track of each fish across their lifetime. Thus, the task of addressing welfare at the level of the individual becomes increasingly problematic. Second, hundreds of diverse species are currently farmed, the vast majority being undomesticated or newly domesticated [29]. There is adaptive between-species variation in many traits that are important for welfare, such as diet and social behavior, with this information being recently organized into an online database called FishEthoBase, yet general species-specific welfare information is available for just 84 of the at least 408 species farmed [2,30]. In addition, such traits typically differ depending on life stage and environment [31,32], and, as outlined above, welfare can vary with individual personality, details of which are known for a small subset of farmed species. Such between- and within-species variation makes designing appropriate environments for all farmed species at all stages a complicated and challenging task. Finally, the industry is relatively young and, as in the case of the terrestrial animal agriculture industry, has historically emphasized health and production traits when considering welfare. Currently, fish welfare scientists recognize that welfare must go beyond simply mitigating health problems to include provision of environments designed to improve other aspects of welfare [33,34,35,36,37,38,39], with some even advocating for provision of environments conducive to positive welfare [25]. Positive welfare is a concept that is still developing in the fish welfare discipline, but in short, goes beyond the Five Freedoms to provision of experiences where agency, choice, control, cognitive stimulation, meaning, and challenge are crucial dimensions to consider [21]. Given the huge number and diversity of fish species being farmed, it is no surprise that this concept has only recently been applied to fishes [25]. The study of what fish need and want in these complex respects, as species, groups or individuals, is still in its infancy [21].

The purpose of this paper is to review the literature that identifies fishes as individuals, examine current fish welfare assessment techniques and burgeoning technologies that can inform the development of more individual-focused assessment, compare those with currently available individual-focused terrestrial animal welfare assessment and monitoring techniques, and make recommendations regarding research avenues that could lead to progress with understanding how to provide each individual fish with conditions conducive to good welfare in captivity.

## 2. Fishes Are Individuals: Implications for Welfare in an Aquaculture Setting

The research into individuality in fishes shows that there is individual variation in cognitive capacities, emotion, and preferences, all of which are linked back to personality [40]. While some of the studies explored throughout this section focused on the individual, others regarded individual variation as noise. This brief review serves not only to highlight that fishes are individuals, thus necessitating welfare assessment for all individuals in a population, but also explores the potential impact of individuality on welfare and the associated difficulties with providing inputs that allow for good welfare outputs for the broad spectrum of individuals in a population.

### 2.1. Personalities and Individual Behavioral Variation

Human psychology has long studied individual differences in behavior that are repeatable across time and across contexts [41,42], or personalities. Animal scientists also began to pay attention to such variability in the 1970s [43], recognizing that individual nonhuman animals also exhibit differences in behavior that are repeatable across time and contexts. Since then, this field of research has flourished [40,44,45,46,47,48,49]. Consistent individual differences in behavior have been characterized as animal personalities [50] or temperaments [51]. When personality traits are correlated with one another, they are called behavioral syndromes [52] and, when associated with differences in physiology, are referred to as stress coping styles [3,4,53,54]. Regardless of the favored nomenclature [55], the important point is that such consistent differences are widespread and strongly impact individual animal welfare [40,56]. For this paper, we will use the terms personality or coping style (depending on the cited author’s preference) to refer to consistent individual differences in behavior and will use the term individual behavioral variation when the authors did not explicitly test for repeatability across time and/or contexts.

Researchers exploring fish personality and individual behavioral variation have examined the behavior of individual fishes with regard to aggression [57], the proactive-reactive coping styles [54], and the bold-shy axis [58,59,60,61]. Fishes’ personality traits intersect with the way they cope with their environments [3] and hence, impact their welfare under captive conditions [62,63] For instance, in Atlantic salmon kept at high density, the personality traits aggressiveness and risk-taking, or boldness, are related and impact fish welfare [63]. Indeed, research has shown that less aggressive individuals secured less food, grew more poorly, and had more fin erosion than more aggressive individuals under unpredictable feed delivery schedules [63,64]. The welfare impacts of a reactive coping style depend on the environmental conditions at the time, as behavioral inhibition can be an adaptive coping strategy during chronic, unpredictable, or uncontrollable situations. It is clear that the threshold for when a stressor shifts from inhibitory to one that warrants proactive responses like aggression is individual in nature [54,65]. The aquaculture industry is attempting to accommodate for variation in personality and behavior and the resulting individual welfare needs by, for example, developing and using demand feeding devices [66,67,68]. While these have had varying levels of success with regard to improving welfare, with that success dependent on multiple parameters including density, social organization, genetics, individual learning ability, and boldness [69], future efforts like these are integral to providing conditions conducive to good welfare for the variety of individuals in each population. 

More research into the impacts of individual behavioral variation on welfare should aim to find creative solutions to allow for good welfare for a diverse range of individuals [70], taking into account potential confounding factors such as group composition, stocking density, and details of the experimental environment. In one study of Nile tilapia, for instance, provision of river pebbles and artificial kelp promoted territorial aggression, thus potentially benefiting successful fighters, while possibly reducing welfare for those who lost fights [71]. However, in a later study, provision of artificial water hyacinths increased aggressive interactions but also reduced stress, on average, as indicated by decreased ventilatory frequency and decreased abnormal repetitive behavior (i.e., scratching) [72]. Because this study did not report results separately for winners and losers of fights, it is unclear how individual behavioral variation impacted the experience in this environment. While neither of these studies investigated whether the ability to win fights was repeatable across time and contexts (i.e., a personality trait), they and others like them highlight the complexity of developing captive conditions that promote welfare over the full range of individuals. Attention should be given to ensuring that a range of environments is offered to accommodate the variety of individuals in the population.

We will briefly explore the implications of individual differences in cognition on welfare in fishes, before examining the resulting preferences and their impact on individual welfare.

### 2.2. Cognition

Much of the research on cognitive capacities in fishes comes out of the discipline of comparative psychology and aims to uncover species-level cognitive capacities. However, such studies also uncover individual variation in cognitive capacity, reflected in the confidence intervals and standard deviations for variables collected in standardized conditions. The relationships between variation in cognition and personality and the implications for the welfare of the individual animals, including fishes, concerned has been well discussed [56,73,74,75,76,77].

While less is known about cognition in fishes than in mammals, the field is growing. Fish are capable of all the main kinds of learning identified in mammals [78,79,80,81,82,83], with individual differences in learning capacity [74,84,85,86,87] and possible relationships with personality. Looking at the relationship between the shy-bold continuum and learning, research has shown, for example, that shy brook trout were better at learning to navigate a maze than were bold brook trout [88], that bold sea trout were better at learning to avoid parasites [89], and that bold zebrafish and guppies showed better inhibition in a tube task [90]. Learning speed has been linked to personality and positively associated with resilience and welfare in farmed animals [91,92]. For example, one study found that bold female guppies learned a new spatial-associative task more quickly and accurately than those who were shy [93]. Given that guppies raised in high predation environments have been shown to be less bold [94], that study confirms the results of a previous study that found that guppies raised in high predation environments were slower to learn [87]. Conversely, once they have learned to cope in one environment, proactive rainbow trout (who are also typically bold) are slower to adjust to environments that have changed [65]. Concerning the physiological aspect of personality differences, rainbow trout who were implanted with cortisol experienced impaired learning, suggesting that individual differences in stress responsiveness could impact learning [95]. Indeed, stressful conditions have been shown to negatively impact learning ability and reaction to novelty [83,96]. It is clear that individual variation in cognition and personality can impact the welfare of individual farmed fish. Understanding the range of personalities present in a population of fishes, how those personalities interact with the range of cognitive opportunities present in the environment, and the variation in how individuals cope with stress [3] is essential to providing all fishes with environments conducive to good welfare. 

### 2.3. Emotion

Animal welfare scientists have known for decades that the capacity to feel emotion is central to animal welfare [20,39,97,98,99,100,101]. Emotions are brief affective responses to an eliciting event, and include an expressive component, a physiological component, and oftentimes a subjective “feeling” experience [102,103]. From an operational viewpoint, emotions have been defined as states elicited by rewards and punishers, that is, by instrumental reinforcers [104]. These states are encoded by the activity of neural circuits that give rise, in a causal sense, to externally observable behavior, as well as to associated cognitive, somatic, and physiological responses [105]. In nonverbal animals, the existence of feelings must therefore be inferred based on the presence of behavioral and physiological responses to an event. Our knowledge of the capacity of animals to experience emotion is much less extensive for fish than for mammals, but research in this area accelerated after publication of research by Victoria Braithwaite and colleagues on pain in fish, with several authors suggesting that fear is a salient emotion in fishes [106,107,108,109,110,111]. More recently, carefully controlled studies on sea bream showed that the model of core emotions based on the two dimensions of valence and arousal correctly predicts behavioral and physiological responses of these fish to positive and negative experiences, indicating that they experience, according to the authors, ‘emotion-like’ states [112]. Indeed, recent research has validated the use of judgment bias testing to measure emotion in zebrafish with the potential to use this as a tool to assess and improve welfare [113]. Together, these various studies suggest that cognition, emotion, and personality are linked and through interactions with the environment, impact the welfare of each individual [56,114].

### 2.4. Preferences

In fish as in other animals, the behavioral and physiological traits that characterize every individual may be linked to emotions that induce cognitive bias [10,115] and in turn, can result in different preferences in many contexts, including the environments they choose to live in, the activities they engage in, and the social partners they affiliate with. Knowledge of such preferences could help to identify and allow for the needs and wants of fish with different personalities held in farming systems. For instance, thermal preference in Nile tilapia seems to be linked to personality, with proactive fishes choosing higher temperatures [116]. Further research using preference testing could explore the needs and wants of proactive and reactive Nile tilapia, thus enabling provision of appropriate resources in each thermal gradient. Experimental efforts like these are best designed when informed by observational pieces of evidence both on wild [117] and, when the opportunity presents itself, farmed conspecifics interacting in their species’ natural habitats. For instance, while juvenile Atlantic salmon typically migrate through fjords into the ocean, some evidence suggests that sexually maturing farmed post-smolts tend to remain in coastal areas and enter rivers as they migrate [118]. Thus, when designing preference studies on potential environmental inputs, it is ideal to look to this type of literature to better understand naturally motivated behaviors and the potential environmental components that allow for those behaviors. Importantly, preference studies must be performed to investigate the range of individual preferences present in a species, or population, and thus inform the inputs in any given captive setting. 

Scientists have performed preference testing in farmed fish species for decades, with work focusing on a range of inputs including plants, enclosure color, temperature, presence of conspecifics, shelters, and food preferences [35,36,119,120,121,122,123]. While some of this research has focused on individual preferences [35,36,119,120,121,122,123], much attempted to determine what the *majority* of individuals within a species prefer. Nile tilapia color preferences have been well studied at the species level, with individual variation revealed in each study [5,124,125]. Investigating decision making and preferences in fish is complex and turning the resulting knowledge into effective tools for improving the welfare of these animals is a laborious process that requires clever experimental design and replicability.

The literature on preferences in fishes is growing but is focused on subset of the 408 species currently farmed [2] and is limited to the choices we have conceived to offer. We do not understand how to ensure high welfare for all of these distinct species farmed [2,21,25], which leaves us very far from knowing how to create environments that provide for the diversity of individual needs and preferences present in any population, but especially in the large numbers characteristic of aquaculture.

## 3. Fish Welfare Assessment Focuses on the Shoal

There are several pioneering studies that explore the topic of fish welfare assessment, with many focusing primarily on inputs designed to result in pre-determined group level, health-related outputs such as body condition, fin condition, and disease status [126,127,128,129,130]. However, as fish behavior has been recognized as an essential indicator of emotion and hence, welfare [114], more recent research has focused on assessing environmental preferences as part of welfare assessment [131]. Very recent fish welfare assessment research highlights the need to include behavioral indicators of welfare, with an awareness of the emotional needs of fishes, but still falls short of measuring welfare at the level of the individual [132,133]. There is also an awareness that behavioral indicators need to be species-specific, with a paucity of research to inform such indicators for most species used in aquaculture [2,21,70]. However, because (1) health is essential to welfare, (2) behavior is often hard to quickly and easily measure underwater, and (3) the goals of the aquaculture industry are to maximize growth rates and minimize production costs [134], indicators of health have been prioritized as cornerstones of fish welfare [126,135]. As this changes and behavioral welfare indicators are included more often, and as research results inform individualized welfare inputs and assessment, the aquaculture industry should prioritize assessment of welfare at the level of the individual, with technology being one tool to do so.

Despite the hurdles to monitoring and ensuring welfare in aquaculture settings, fish welfare scientists have been exploring the use of technology as a replacement for the human–animal relationships that historically formed naturally when animals were farmed in smaller numbers, thus informing animal care, but are impossible in large-scale aquaculture with millions of individuals [136]. Similar to the difficulties associated with forming millions of human–fish relationships, individual welfare appears difficult to assess in these large group numbers. However, there are several burgeoning technologies available that, if adapted properly, could inform progress toward this goal. Precision Fish Farming (PFF) aims to, among other goals, facilitate autonomous and continuous animal monitoring, and as a result, improve animal health [136]. Submerged cameras are commonly used to observe and analyze fish behavior, with more sophisticated computer vision methods [137], including machine vision systems, also available to measure behavior, especially related to feeding [136,138,139,140,141,142,143,144,145]. Technical difficulties arise and solutions are being studied to prevent occlusion that occurs when using machine vision systems to collect behavioral data on groups of fish where multiple fish are close enough together to be interpreted as one [141]. Hydroacoustic devices have also been employed to collect data on schooling behavior [146], with more precise split beam sonars capable of measuring swimming behavior in individual fishes [147,148,149]. Hydrophones have been used to record sounds emitted by the fishes [150,151]. Acoustic fish telemetry has been used to monitor individual wild fish behavior and adapted to an aquaculture setting to explore swimming activity and depth and physiological measures including respiration rate and feed intake [136,152]. While theoretically this could be used to monitor and track individual welfare over time, this technology is highly invasive, requiring surgery to implant the sensor and thus potentially impacting the variables of interest [153]. Additionally, in the high-density situations characteristic of fish farming, it is likely that the number of tags would overload the receivers. This hurdle, along with associated increased mortality [154], reduces the utility of tagging in tracking large numbers of individuals characteristic of aquaculture [136]. Passive integrated transponder (PIT) and bio-sensor technology may be feasible alternatives, but also carry potential welfare costs [154,155,156].

### Why Don’t Group Level Assessments Ensure High Welfare for All Individuals?

Group level welfare assessment techniques sample a small percentage of the individuals in a group, extrapolating that information to provide an average welfare metric for a shoal. For example, the 2014 version of the RSPCA’s welfare standards for farmed rainbow trout included auditable animal welfare outcomes (e.g., fin damage, operculae damage, and eye damage) to be measured in a sample of 100 fish at the point of killing, with a goal that no more than 10% of fishes score above 1- indicating damage (it should be noted that these standards were not included as auditable in the 2020 version, but may be reworked for future iterations) [157]. When stated in this way, it becomes clear that the goal at slaughter is not even to measure health and welfare in all individuals, let alone provide environments conducive to good welfare for all individuals, but instead to measure the percentage of individuals who are experiencing poor welfare, and ensure that number does not exceed 10%. Considering that the rainbow trout industry slaughters at least 87 million trout per year [28], this means that upwards of 870,000 rainbow trout per year experience poor welfare. Furthermore, this strategy relies on the likely untrue assumption that the 100 fishes who are sampled are statistically representative of the tens of thousands of trout on a farm [158].

While these group-level metrics can be useful in implementing changes that might prevent the worst suffering for some portion of future populations, there are undoubtedly individual animals who are outliers and are not meeting the welfare targets, even if the average metric indicates that this is indeed the case for most animals within that population [159,160]. Consider, hypothetically, a normally distributed bell curve where 15.8% of the population is more than one standard deviation below the mean. If the mean value were able to indicate a certain welfare level, then in a population of 100,000 fishes, at least 15,800 fishes would not meet the welfare level set as a goal. Extrapolate that to the estimated number of fishes raised worldwide, and over 15 billion fishes, all individual animals, would experience welfare below that which the industry aims to provide.

More research must be done to better understand the needs and wants of individual fishes, and thus inform existing protocols like these and allow for the integration of emerging technologies to inform individual welfare assessment. In order to make progress with understanding the needs of farmed animals and ways to assess their individual welfare, we can look to facilities where animals live their lives in group numbers small enough to allow development of scientifically validated individual animal welfare assessment protocols. Even more important, these efforts are instrumental in further understanding which conditions are compatible with providing individual animals with the highest possible levels of welfare.

## 4. Can We Further Take Inter-Individual Differences into Account in Animal Welfare Assessment?

### Zoos Utilize Individual-Focused Welfare Assessments

Taking a cue from the animal agriculture industry [161], the zoo animal welfare field has, in recent years, begun to formally recognize the importance of animal welfare and outlined guidelines for both animal welfare research and assessment [162,163,164,165,166]. While many zoos recognize the usefulness of epidemiological studies that aim to improve the health and welfare of the population [167,168], there is also a recognition that factors promoting welfare for some individuals do not benefit others and hence, more progressive zoos have developed welfare solutions tailored to individual animals [166,169,170,171,172,173,174]. Likely because zoos are assessing welfare in small populations of, for example, a few gorillas or a few dozen penguins, the importance of the individual as the unit of study is now highlighted throughout the zoo animal welfare assessment literature and is usually realistic to carry out [13,165,175,176,177]. Assessments utilize multiple animal-based measures for every individual (e.g., behavioral and physiological) over an individual’s entire life span [166] in addition to ensuring the appropriate resource inputs (e.g., sufficient space, appropriate and preferred diet, provision of enrichment to elicit natural behaviors, etc.) [13].

There is a wide range of assessment techniques available to zoo professionals, from holistic assessments that take into account measures of health, emotion, and behavior but require a high level of resources, time, and technical skills [178,179,180,181] to techniques that are less resource intensive but can be used by zookeepers to monitor welfare more regularly [182,183,184]. Although many of these tools are similar to and may even have originated in the animal farming industry, because of the zoo focus on individuals, they are used differently and thus, the aquaculture industry can look to zoos for ways to measure welfare at the level of the individual. While techniques have been assessed and validated that use glucocorticoid and behavioral data to assess an animal’s welfare state in ways that are familiar to fish welfare scientists [185,186,187,188], experienced zoo animal caretakers who have cultivated relationships [189] with the individual animals may be able to gain additional knowledge through a more intuitive approach including the observation of very minor changes in an individual animal’s behavior, posture, eating habits, and movement patterns [13,190,191,192]. While these welfare indicators are sometimes used at the level of the population in aquaculture settings, the current feasibility of employing such approaches at the level of the individual is very low.

Integrating the goals of individual welfare assessment tools with emerging technology (see Section 3) could allow for the creation of innovative tools that make progress towards providing good welfare for all individuals. Indeed, zoo animal welfare researchers have recently investigated the use of emerging technologies, many coming from research in farms and laboratories, as one strategy to monitor welfare in elusive, cryptic, and nocturnal individuals, with the goal being to explore an animal’s behavior and/or affective state [175]. These technologies include the use of accelerometers, global positioning systems, and radio frequency identification systems to investigate movement patterns and even lend insight into social relationships of individuals, a potentially important indicator of fish welfare, especially in territorial species [175,193,194,195,196]. Bioacoustics have been used to investigate emotion and hence, welfare, through analysis of vocalizations [25,197]. Of course, this would be difficult at high stocking densities, however, investigating the use of bioacoustics in experimental settings may lend insight into ways to adapt this technology to commercial settings. Additionally, technology can be used to assess the level of environmental factors, like sound, with the goal of understanding how they can impact individual welfare [198]. Creatively combining the holistic, individualized, lifelong approach to monitoring welfare seen in the zoo industry with these technologies may allow fish welfare scientists to uncover the wide range of welfare states in a population, adjust inputs to accommodate the individual variation, and iteratively measure welfare.

In addition to more objective measures of welfare, evidence that emerges from human–animal relationships can inform how individual welfare is measured. In zoos, caretaker opinions of welfare have proven to be useful indicators of welfare [182,183,199]. Fish welfare researchers could take this a step further by collaborating with social scientists to utilize multispecies ethnography [200,201] as a way to determine which indicators of welfare are intuitively extracted from the human–animal relationships that occur at the individual level in zoo settings, and work with engineers to adapt technology to measure those indicators across an entire population of fishes.

## 5. Conclusions: Making Progress with Understanding How to Provide Fishes with Good Welfare at the Individual Level under Captive Conditions

Providing conditions conducive to individual positive welfare for the tens of billions of farmed fishes comprising hundreds of species raised in a variety of systems represents a formidable challenge. Historically, animal welfare scientists have examined behavior in wild species to understand which behaviors are essential and to inform experimental research exploring the upper limits of animal welfare in captive conditions [202]. This approach becomes even more useful when investigating the needs of farmed fish species which have undergone less domestication than terrestrial farmed species and thus are more likely to have similar needs to their wild counterparts. However, the fact that these species live underwater, often with life stages occurring in distinct aquatic environments, means that progress is slow on understanding the behavioral needs and preferences of the numerous species farmed in aquaculture, and hence we have not yet explored the current upper limit for welfare in different farmed fish species [2,21]. As we have seen, there is a growing body of research informing fish welfare. To push the current upper boundaries of individual welfare in captivity, however, researchers should place an increased emphasis on testing conditions that exceed conventional aquaculture settings with the goal of revealing what captive fishes truly need to thrive. Creative research has the potential to progressively uncover individual behavioral needs and preferences that would otherwise go unknown in aquaculture, (1) highlighting areas where improvement is needed; (2) flagging some settings as unable to provide high welfare; and (3) revealing that, when given appropriate social and physical environments as well as agency and meaningful choices, the level of welfare experienced by captive fishes can far surpass that which is currently provided by the aquaculture industry. In turn, this research can inform individualized welfare assessments that could further take advantage of developing technologies like precision livestock farming to literally measure and track welfare in every individual.

The importance of ensuring individual high welfare in captive fishes, while recognized by fish welfare scientists, is currently in conflict with the financial goals of the aquaculture industry and hampered by the relative paucity of scientific knowledge. As we continue to tackle this challenge, we might find that for some species or life stages the target of individual high welfare in captivity is simply not possible or profitable. Yet, the number of fishes who are being farmed is only growing, which means that the number of individuals who are being left behind by welfare assessments that largely utilize group-level welfare indicators is also growing. There is an urgent need to make progress with identifying the inputs necessary to provide good welfare for the variety of individuals who are kept in captive conditions around the world.

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
