# Peer review of "Looking beyond the Shoal: Fish Welfare as an Individual Attribute"

_animals, 2022, doi:10.3390/ani12192592_

Round 1
Reviewer 1 Report
Review: Looking Beyond the Shoal Fish Welfare as an Individual Attribute
This manuscript is a review aiming to build a case for fish as individuals, and examining the necessity of considering fish welfare in the aquaculture field at the individual level. The review is well written throughout, and the topic is timely for the field. My comments are overall relatively minor points for clarification or restructuring, and I hope to see this paper in publication soon.
One specific concern I have throughout the manuscript is that the authors need to be extremely explicit with their definitions of various terms, as the animal welfare field is already fraught with examples of multiple definitions of animal welfare being used interchangeably. In the introduction, the authors introduce the phrase “positive welfare” and cite broadly accepted experiences important to providing positive well-being in many vertebrates – agency, choice, control, challenge, meaning etc (Line 83). No statement is made as to whether this is what the authors are using as their definition of welfare through out the rest of the manuscript, and the definition of welfare referenced through various citation seems to drift between emotional/affective based definitions to coping with stressor definitions to physical health.
For instance –
Line 114 – Authors note individual differences impact animal welfare, but there is no indication what definition of welfare is being referred to here
Line 118 – Welfare is equated to coping style of individual animals with stress
Line 123 – “The welfare impacts of a reactive coping style are not fully understood…” – this suggests welfare is something separate from coping strategy that can be influenced by individual coping strategy – please clarify
Line 180 – Another reference to coping strategies, welfare is not explicitly stated here but it is implied to be related to appropriate environments?
Line 202- “Together, these various studies suggest that cognition, emotion, and personality are linked, with implications for the welfare of individuals…” This statement seems to imply cognition, emotion and personality are all separate from the authors’ definition of welfare
I suggest the authors define their position on what welfare actually is for an individual fish in the introduction of the paper. From the manuscript, it seems their definition is the individual’s experience of all aspects of its environment, including the ability to express behaviors in a manner consistent with that individual’s personality, cognitive abilities and preferences.
I think this is an excellent definition of welfare and holding aquaculture to this standard is important. Because of this, I think it is imperative the authors be very clear about what they are referring to as welfare so that the fish welfare field does not “reinvent the wheel” or set itself up to have the same struggle as other welfare fields in how welfare is defined (health? Mental wellbeing? Behavioral expressiveness? Etc). From there, the points about individuality and its impact on welfare will be much clearer, as the authors can reference how individual differences will impact the specific form of welfare they are discussing.
Similarly, “personality” at times in the manuscript seems to be as a catch-all term for any variation among individuals, when in several cases these differences may not be entirely attributable to consistently reproducible behavioral differences.
For instance –
Line 97- “…all of which are linked back to personality, which is individual by definition.” This is correct in that individuals all have unique personalities, but in the academic sense of the word personality traits are measured within a population and the relative degree to which individuals express certain traits is the measure of interest. Suggest deleting the clause “which is individual by definition” here, technically speaking personality cannot be measured in a single animal without knowing the range of specific traits in other individuals of the species.
Line 138 – “In one study of Nile tilapia…” This statement fails to connect personality traits to territorial behaviors. Winning fights or losing fights are not personality traits, and shy individuals may still be motivated to obtain and defend territories. Moreover, the provision of a more complex environment may not be a confounding factor when considering personalities, as more environmental variation may allow for differential expression of personality traits which seems to be an opportunity the authors wish to promote in farmed fish.
Line 144 – “..Because this study did not report result separately for winners and losers of fights, it is unclear how different personality types experienced this environment.” This statement makes the large assumption that only bold individuals win and only shy individuals lose, whereas I would imagine bold individuals may across the board be more willing to engage in conflict and therefore are more likely to both win and lose when compared to shyer individuals. Again, winning and losing fights are not personality traits.
Line 170 – “confirming the results of a previous study that found guppies raised in a high predation environments were slower to learn.” – This does not necessarily follow, being raised in a high predation environment seems more related to stress levels and early life experience than personality. The cited paper here does mention personality traits, so consider referencing those traits instead of the predation environment.
Lines 206-211 – Line 210 specifically seems to suggest all of the previously stated variations among individuals may be directly linked to personality when most likely there are multiple etiologies leading to individual preferences in environments and social partners.
My take away is the authors intend to highlight the individuality of fish which is a worthwhile point, but instead reference ‘personality’ as a stand-in for ‘individuality’ and the two are not equivalent. This does not negate the value of the paper, however the paper would be much clearer with an outright statement of what definition the authors are using for personality, and references to the specific personality traits measured in the cited papers to verify they are indeed discussing personality differences. If there are no specifically measured personality traits, consider replacing “personality” with “individuality” or “individual variation”.
Minor comments
Line 50 – “in response to the same challenge” – consider changing challenge to “condition” or “event”
Line 51 – “may show up as” – change to “may be expressed as”
Line 85 – Break up this sentence, end after citation [32]
Line 109 – This is a long sentence, break up, end after citation [45].
Line 125 – Long sentence, break up after “uncontrollable situations”
Line 127 – “The industry” – change to “The aquaculture industry”
Line 130 – “and sometimes negatively impact welfare” – this is an odd aside and does not seem well supported? Consider putting this as its own sentence with a citation/explanation or removing
Line 135 -148 – This paragraph is odd in that the authors almost seem to be advocating for less complex environments due to additional environmental features stimulating more perceived negative behaviors in fish? I would imagine this is the opposite of what would benefit farmed fish – more complex environments seem to be beneficial for other vertebrates across the board. The effect observed in these studies may be an effect of observing a wider variety of behaviors simply because the fish now have the opportunity to express more behaviors due to increased environmental complexity. Consider adding a statement to this effect, it seems detrimental to suggest more barren environments are desirable.
Line 226-234 – The point about preference testing is a good one, it is worth acknowledging the limitations of it though. Specifically, we can only learn about preferences the species have based on what we choose to present to them. There are likely many things they might prefer to have or not have that we are unaware of and simply do not offer, so they are limited to expressing preferences within the bounds of what we can conceive of.
I wonder if section 3 may fit at better before section 2? Start from where welfare assessments are now (shoal/school level) and lay out the arguments for why fish need to be considered as individuals.
Section 4 is generally well written but many of the zoo references are over 10 years old. A lot of research in relevant areas has come out of zoos in this time period, please consider updating this section more.
Line 396 – “upper limit is for welfare” – this is an odd phrase, in other animals we seem to accept that we improve welfare and this become the ‘new normal’ to which the animal adjusts, so we seek to continuously improve. This isn’t because there is an absolute welfare level the animals are capable of experience, but because reward systems are adaptable. Rephrase please.
References
Throughout the references they vary between using journal abbreviations and using full journal names, check what the requirement is for the citation style and make sure they are consistent.
Some Animals citations include (Basel) in the title of the journal which I do not think is correct
Some citations include DOIs and some do not, check citation style requirements
25 – This appears to be a thesis and is cited incorrectly
34 – This appears to be a book citation but is formatted incorrectly
Whitham and Wielebnowski is cited twice – 177 and 194
Reviewer 2 Report
Overview: This is an exceptional paper that not only provides deep insights into existing literature on fish behaviors and cognition, but also a thorough background into the meaning of animal welfare at both a group and individual level and the biases that often interfere with the implementation of better management in different industries. This paper creates a strong argument for fish welfare and is also highly relevant to other areas of animal welfare both in and out of agricultural environments.
Abstract: Very clear as to the history of the topic and the nature of the argument for individual fish welfare given the current context of fisheries and fish farming.
Introduction:
Line 48 – consider adding a comma after “population”
Line 77 – consider adding in a reference for the sentence that discusses how agriculture has emphasized welfare with respect to production traits. There are far too many studies to cite them all, but perhaps finding a few reviews would work or citing a different source.
Fishes as Individuals: Very thorough and well-written. No suggested changes.
Despite their individual needs….:
Consider shortening the title of this section
It was good to see an overview of the available technologies that are often used in different environments and assessing their compatibility with fisheries and attention to welfare.
Can we further take…:
This section does a great job of contextualizing the field of animal welfare as it has progressed from the agricultural world into zoos (where more individual focus is done) and seeing if this approach can be applied to fisheries.
Making progress with understanding…:
The authors do a great job of addressing the underlying issues of supporting animal welfare in fisheries and why this is such a challenge.
